# Score-Based Generative Models for Wireless Channel Modeling and Estimation

**Marius Arvinte, Jonathan I. Tamir**
Electrical and Computer Engineering
The University of Texas at Austin
`{arvinte,jtamir}@utexas.edu`

## Abstract

In this work, we investigate score-based models for learning the distribution of multiple-input multiple-output (MIMO) wireless channels in structured stochastic environments, using either clean or corrupted (noisy) data for training. We find that score-based models are capable of generating high-quality synthetic channels, and have robust downstream estimation performance, sometimes surpassing strong baselines by up to 10 dB in estimation error, when the inverse problem is ill-posed. Our preliminary results on training with corrupted data show improved performance against simple baselines, and introduce a very promising future research direction. Code is publicly available at `https://github.com/utcsilab/score-based-channels`.

## 1 Introduction

The field of digital communications is a backbone of our global society, and, in fact, has effects in modern deep learning – as in the case of federated learning, where the communication budget is a key motivator and constraint (Kairouz et al., 2021). A deciding factor for the end-to-end performance of a communication system is given by the statistics of the wireless environment, summarized by a *channel model*. Proper channel modeling is thus crucial for downstream tasks, such as channel estimation (Li et al., 2002), beamforming (Sun et al., 2014), and data recovery (Weber et al., 2006).

Deep learning has recently been successfully applied to several of these signal processing tasks in isolation. However, a generative model for wireless channels that can offer high performance on multiple downstream tasks remains an open problem. Furthermore, a particular challenge in deep learning for communications comes from an *imperfect data* problem: all acquired measurements suffer from corruptions due to hardware and resource limitations, even in controlled environments (Molisch et al., 2016; O'Shea & Hoydis, 2017). From a learning perspective, this is equivalent to the problem of learning generative models of the underlying distribution using only noisy, and potentially undersampled, linear measurements (Bora et al., 2018; Lehtinen et al., 2018).

Score-based generative models (Song & Ermon, 2019; 2020; Song et al., 2021) have recently emerged as a powerful modeling tool for highly structured data distributions. In this work, we use score-based models to learn the distribution of wireless channels generated from standardized stochastic models used across the cellular industry (3GPP, 2020). We resort to simulated channels because of a lack of publicly available real-world, high-dimensional measurement datasets, but also due to the widespread acceptance of these models.

We build on the method in Arvinte & Tamir (2021) and more broadly investigate the use of score-based generative models for MIMO wireless channels. Our contributions are: (i) we evaluate the sampling quality of score-based models and generative adversarial networks (GANs) trained on simulated environments, (ii) we perform comparisons of score-based models with an extensive range of baselines when applied to ill-posed MIMO channel estimation as a downstream task, and, (iii) we propose a Stein's unbiased risk estimate (SURE) loss formulation for score-based models when only corrupted data are available for training, and compare this to naive (unmodified) training of score-based models with noisy data. Our results on channel estimation show that score-based models are an extremely attractive and robust approach, in some cases surpassing even algorithms that exploit knowledge of the underlying stochastic model used to simulate the environment.

## 1.1 RELATED WORK

The work in Balevi et al. (2020) is an application of the compressed sensing with generative models (CSGM) framework (Bora et al., 2017) to wireless channel estimation. This method trains a GAN with a low-dimensional latent space, that is then optimized in this latent space in conjunction with the observed measurements during inference. The generative latent optimization (GLO) approach Bojanowski et al. (2018) learns a latent codebook for the training set instead of using an adversarial approach, and can be used in an identical way during inference.

Using conditional generative models to learn the effects of unknown channels has been proposed in O'Shea et al. (2019), but this method cannot explicitly output channel realizations. Compressed sensing has been used for MIMO channel estimation, based on a sparse channel model (Saleh & Valenzuela, 1987). The Lasso is a fast estimation method (Schniter & Sayeed, 2014), while the approximate atomic norm decomposition (fsAD) (Zhang et al., 2017) imposes sparsity in an over-sampled representation, and exploits the underlying clustered delay line (CDL) channel models.

End-to-end training of unrolled optimization approaches have been successfuly used for wireless channel estimation in He et al. (2018) and offer a strong and robust deep learning baseline in the form of the learned denoising AMP (L-DAMP) algorithm. Finally, recent work in Kim & Ye (2021) proposes learning a denoising model using corrupted data at a single noise level.

## 2 SCORE-BASED GENERATIVE MODELS FOR WIRELESS CHANNELS

We consider the setup of narrowband, MIMO communications (Tse & Viswanath, 2005), even though the framework of score-based models is extendable to higher-dimensional wireless channels. In this setup, a channel realization is represented by a complex-valued matrix $\boldsymbol{H} \in \mathbb{C}^{N_\mathrm{r} \times N_\mathrm{t}}$ sampled from an underlying distribution $p(\boldsymbol{H})$, where we assume a transmitter and receiver each equipped with $N_\mathrm{t}$ and $N_\mathrm{r}$ antennas, respectively. Channel estimation consists in recovering $\boldsymbol{H}$ from a set of $N_\mathrm{p}$ pilot transmissions, typically staggered across time (Larsson et al., 2014), and known ahead of the time by the receiver. Each transmitted, unique, pilot vector $\boldsymbol{p}_i \in \mathbb{C}^{N_\mathrm{t}}$ is seen by the receiver as $\boldsymbol{y}_i = \boldsymbol{H}\boldsymbol{p}_i + \boldsymbol{n}_i$, where the channel imposes a linear effect on the pilots, and adds complex-valued Gaussian noise $\boldsymbol{n}_i$ with zero mean and variance $\sigma_n^2$. Assuming that the channel and noise variance are constant across all $N_\mathrm{p}$ pilot transmissions, the receiver observes the matrix of measurements:

$$\boldsymbol{Y} = \boldsymbol{H}\boldsymbol{P} + \boldsymbol{N}. \tag{1}$$

In practical systems, the elements of $\boldsymbol{P}$ are commonly restricted to unit amplitude to limit power consumption and nonlinear effects. The signal-to-noise-ratio (SNR) is defined as $N_\mathrm{t}\mathbb{E}[|h_{i,j}|^2]/\sigma_n^2$. When the pilot density $\alpha = N_\mathrm{p}/N_\mathrm{t}$ is strictly less than one, the above problem takes the form of an under-determined inverse problem, and knowledge of the prior $p(\boldsymbol{H})$ is needed to successfully recover $\boldsymbol{H}$ from $\boldsymbol{Y}$. An important characterization of a channel model is given by the distribution of its instantaneous capacity (Goldsmith et al., 2003) $C = \log_2 \det \left(\boldsymbol{I} + \boldsymbol{H}\boldsymbol{H}^H\right)$, measured in bits per channel use, where $\boldsymbol{I}$ is an identity matrix.

We learn a noise-conditional score network (Song & Ermon, 2020) using a training set of 10000 ideally known channels from the CDL-C MIMO channel model (3GPP, 2020), with $N_\mathrm{t} = 64$ and $N_\mathrm{r} = 16$. We treat the real and imaginary parts as separate channels and normalize $\boldsymbol{H}$ during training and testing by division with $\mathbb{E}_\mathrm{train}[|h_{i,j}|^2]$. The loss function for the score-based model $s_\theta$ with weights $\theta$ and a batch of $B$ samples takes the form (independent of $\boldsymbol{P}$):

$$\frac{1}{2B} \sum_{i=1}^{B} \left\| \sigma_i s_\theta \left(\boldsymbol{H}_i + \boldsymbol{Z}_i; \sigma_i\right) + \frac{\boldsymbol{Z}_i}{\sigma_i} \right\|_F^2, \tag{2}$$

where, for each step, the elements of $\boldsymbol{Z}_i$ are i.i.d. sampled from $\mathcal{N}(0, \sigma_i^2 \boldsymbol{I})$, and $\sigma_i$ is picked uniformly at random from exponentially spaced noise levels chosen according to (Song & Ermon, 2020). For sampling and inference, we run annealed Langevin dynamics, with the update rule given by:

$$\boldsymbol{H}_{t+1} \leftarrow \boldsymbol{H}_t + \gamma_t \left(\boldsymbol{Y} - \boldsymbol{H}_t\boldsymbol{P}\right) \boldsymbol{P}^H + \eta_t s_\theta \left(\boldsymbol{H}_t; \sigma_t\right) + \sqrt{2\beta\eta_t}\boldsymbol{Z}_t \tag{3}$$

where $\gamma_t, \beta$ and $\eta_t$ are tunable hyper-parameters, and $\sigma_t$ ranges from largest to smallest.

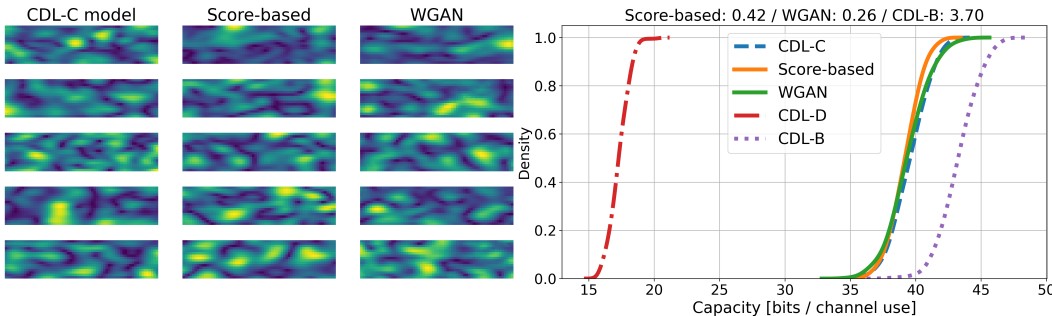

Figure 1: Sampling diversity and similarity with ground truth distribution for score-based and adversarial generative models trained on CDL-C channels. The left columns show five exemplar samples from the ground-truth model and the two models, respectively. The rightmost column shows the empirical capacity CDF for each of the models, and an additional two different environments, with the 1-Wasserstein distance between CDL-C channels and various other models shown in the title.

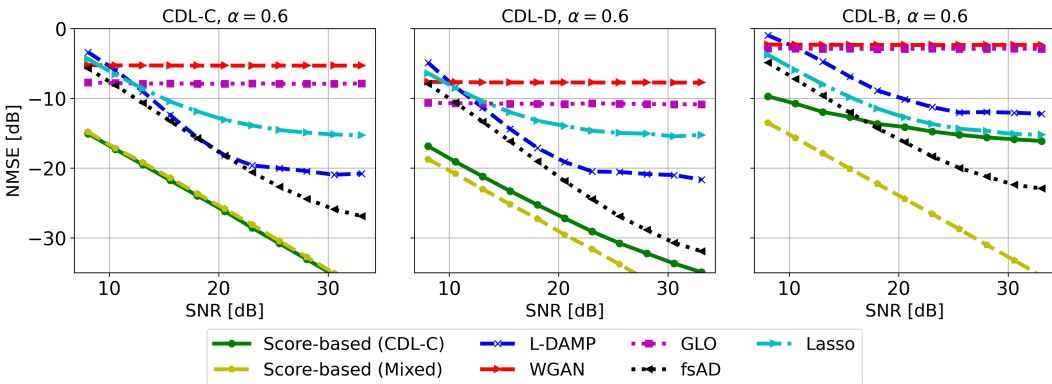

Figure 2: Estimation performance for models trained and tuned on CDL-C (moderate scattering) channels. Except for the model labeled *Score-based (Mixed)*, all other methods are trained and tuned exclusively on CDL-C channels. The left plot shows in-distribution estimation normalized mean squared error (NMSE), while the other plots evaluate generalization to completely different environment models. Score-based models are exceptionally robust in CDL-D (line of sight) environments, and also surpass competing deep learning methods in CDL-B (rich scattering) environments.

## 2.1 Sampling Learned Channels

We sample channels from the learned $p\left(\boldsymbol{H}\right)$ by setting $\gamma_t = 0, \beta = 1$, and we set $\eta_t$ as in (Song & Ermon, 2020). Figure 1 shows random samples from the ground truth CDL-C environment, the score-based model, and the WGAN approach in Balevi et al. (2020), respectively. From qualitative inspection, the sampled channels look similar to true CDL-C channels. Quantitatively, we find that the empirical 1-Wasserstein distance between the capacity of synthetic channels and CDL-C channels (0.42 for score-based models) is much smaller than compared to CDL-D or CDL-B channels (both greater than 3), which is expected as they have different statistics of their modeled environments.

We conclude that both score-based models and GANs can successfully learn the distribution of MIMO wireless channels from standardized stochastic models, and can be a useful tool for generating synthetic data from targeted environments.

## 2.2 Channel Estimation

We perform channel estimation via posterior sampling (Jalal et al., 2021b; Arvinte & Tamir, 2021) by using the trained score-based model in equation 1. The hyper-parameters are tuned using a vali-

dation set of 50 CDL-C channels, regardless of the test-time distribution. We include $\beta$ as a means to dampen the added noise during inference, which we find helps with validation performance. We use a $\boldsymbol{P}$ matrix with randomly sampled entries from $\frac{1}{\sqrt{2}}(\pm 1 \pm 1j)$.

Figure 2 confirms the near-optimality of posterior sampling in-distribution (Jalal et al., 2021b), as well as the distribution-shift robustness (Jalal et al., 2021a). The WGAN and GLO baselines saturate in performance in the high SNR regime Bora et al. (2017). This could be potentially remedied through the use of more expensive inversion methods (Daras et al., 2021). Score-based models also outperform all other baselines, except in CDL-B environments, where fsAD is competitive. When access to multiple environments is available for training the *Mixed* model, score-based models achieve state-of-the-art MIMO channel estimation performance in the considered SNR range.

## 2.3 MULTILEVEL SURE TRAINING OF SCORE-BASED GENERATIVE MODELS

We now assume that we only have access to a training set of corrupted channels, $\boldsymbol{H}_{w,i} = \boldsymbol{H}_i + \boldsymbol{W}_i$, where the noise $\boldsymbol{W}_i$ is i.i.d. Gaussian with zero mean and variance $\sigma_w^2$. Assuming knowledge of $\sigma_w^2$ – practical in wireless systems, where noise level estimation is well understood (Barhumi et al., 2003) – we propose the following training objective for a denoiser $g_\theta$ based on SURE:

$$\frac{1}{B}\sum_{i=1}^{B}\|g_\theta\left(\boldsymbol{H}_{w,i}+\boldsymbol{Z}_i;\sigma_i\right)-\left(\boldsymbol{H}_{w,i}+\boldsymbol{Z}_i\right)\|_F^2+2\left(\sigma_i^2+\sigma_w^2\right)\operatorname{div}_{\boldsymbol{H}_{w,i}+\boldsymbol{Z}_i}g_\theta. \tag{4}$$

We use the Monte-Carlo SURE formulation (Ramani et al., 2008) to approximate the divergence term and learn to remove two corruptions simultaneously: the intrinsic noise $\boldsymbol{W}_i$ at a fixed level $\sigma_w$, and the added noise $\boldsymbol{Z}_i$, at randomly sampled levels $\sigma_i$. During inference, we convert the learned model $g_\theta$ to a score-based model using the functional form $s_\theta\left(x;\sigma_i\right)=\left(g_\theta\left(x;\sigma_i\right)-x\right)/\sigma_i^2$, followed by posterior sampling with (1).

We consider two baselines: score-based models using clean data, and naively trained score-based models using corrupted data. Figure 3 shows estimation results when using $\boldsymbol{P}=\boldsymbol{I}$ and $\sigma_w^2=0.1$ corruption for the train set – note that $\sigma_w^2$ is distinct from the measurement noise during channel estimation, which we vary. It can be noticed that the naive score-based models fail to estimate the channels accurately in the high measurement SNR regime, whereas the proposed approach scales more favourably, but still leaves a performance gap against having a clean training set.

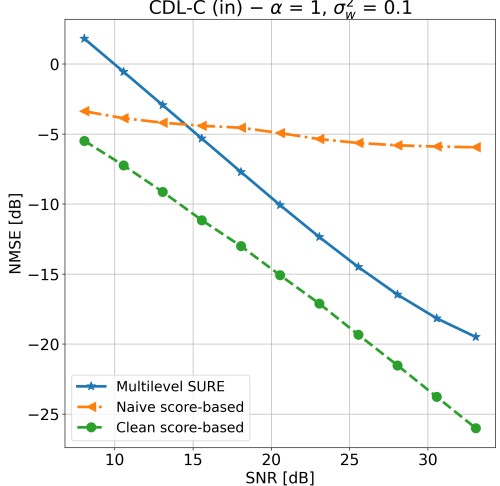

Figure 3: Estimation performance when training using noise-corrupted channel realizations. At high measurement SNR, the proposed multilevel SURE approach surpasses the baseline of naively trained score-based models with noisy data.

## 3 CONCLUSION

We have shown the potential of score-based generative models for wireless channel sampling and estimation, as well as learning directly from corrupted data. Our results were obtained by an almost out-of-the-box usage of score-based models: this validates their representational power, but also leaves future room for important, wireless-specific modifications brought to these models, such as training methods that leverage the beamspace sparsity of MIMO channels or that can better handle corrupted data. Finally, latency is still an open problem – while a score-based model with as few as $500K$ parameters can accurately learn the CDL-C channel model, inference still costs as much as 6 seconds per sample, limiting the current approach to static environments and leaving room to integrate recent techniques that accelerate inference, such as in Salimans & Ho (2022).

## ACKNOWLEDGMENTS

We thank the anonymous reviewers for their useful comments. This work was supported by ONR grant N00014-19-1-2590 and a gift made by InterDigital, an affiliate of the 6G@UT center within the Wireless Networking and Communications Group at The University of Texas at Austin.

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

## A  WIRELESS MIMO CHANNELS

The clustered delay line (CDL) family of channel models are widely used in the wireless industry for simulating the end-to-end performance in different types of environments 3GPP (2020). As their name implies, these models are stochastic and simulate an environment by sampling clusters of propagation paths. In general, the lower letters A, B, C represents richer channels in terms of the number of paths, while D and E are line-of-sight channels with fewer propagation paths. While an increased number of paths benefits the overall capacity, it also makes the environment less sparse (in the continuous angular domain), and consequently channel estimation a harder problem.

At a high-level, these models synthesize channels by first sampling pairs of transmit and receive angles (both in azimuth and elevation directions – here we only consider linear arrays which are only affected by the azimuth direction, for simplicity) $\phi_{\text{t},i}$ and $\phi_{\text{r},i}$, together with clusters of path delays $\tau_i$, path gains $h_i$ and Doppler delays $\nu_i$. Once a set of angular values, delays, gains and shifts are sampled, the four-dimensional channel tensor $\boldsymbol{H}$ between a transmitter and receiver each with $N_{\text{t}}$ and $N_{\text{r}}$ antennas, respectively, is constructed as (Bajwa et al., 2010):

$$\boldsymbol{H}\left(f,t\right) = \sum_{i=1}^{L} g_i \exp\left(-j2\pi\tau_i f\right) \exp\left(j2\pi\nu_i t\right) \boldsymbol{a}_{\text{r}}\left(\phi_{\text{t},i}\right) \boldsymbol{a}_{\text{t}}^{H}\left(\phi_{\text{r},i}\right), \tag{5}$$

where $L$ is the total number of propagation paths, and $\boldsymbol{a}_{\text{t}}$ and $\boldsymbol{a}_{\text{r}}$ are the transmitter and receiver array response functions, respectively. These functions generally depend on the configuration of the antenna array (either analog, or after beamforming) used at the transmitter and the receiver, independent of each other.

At a fixed frequency $f_0$ and time $t_0$, MIMO propagation is characterized by the complex valued channel matrix $\boldsymbol{H}\left(f_0,t_0\right)$. The $(i,j)$-th entry in the channel matrix represents the complex-valued gain (magnitude and phase shift) between the electromagnetic wave field transmitted by the $j$-th antenna, and received by the $i$-th antenna. Thus, for arbitrary channel matrices, a transmitted vector of complex-valued symbols is seen as "entangled" at the receiver side. Note that the forward model in equation 1 also includes Gaussian noise, which is applied at the receiver side (after the channel multiplies the transmitted vectors – in this case, the columns of $\boldsymbol{P}$). This is caused by the thermal noise in the electronic components of the receiver, and modeling with a Gaussian i.i.d. distribution is a well-accepted model Tse & Viswanath (2005).

In this work, we generate MIMO channels by using a frequency $f = 40$ GHz – which corresponds to millimeter-scale wavelengths $\lambda$ – and randomly sample $t$ from a set of ten equally spaced values for each channel realization. We consider uniform linear spaced arrays at both the transmitter and receiver, with an antenna spacing equal to $\lambda/2$. This specific form leads to $\boldsymbol{a}_{\text{t}}$ and $\boldsymbol{a}_{\text{r}}$ both being rows of a discrete Fourier transform matrix of the appropriate size, and the synthesized channels being exactly sparse on the continuum of frequencies. This is the reason why we consider the fsAD approach to be a strong baseline, and why it unsurprisingly surpasses all prior deep learning methods.

In reality, channel sounding work suggests that this exact sparsity is violated (Molisch et al., 2016), but there is currently no public dataset of measured high-dimensional MIMO channels available. This violation of exact sparsity represents one of the main reasons for considering score-based generative models a prime candidate for learning the distribution of wireless channel, since these models, unlike GANs, do not require the explicit definition of a low-dimensional latent space, instead operating in the ambient (high-dimensional) space. Other generative approaches of similar nature, such as normalizing flows (Papamakarios et al., 2021), are interesting research directions to consider.

## B  SCORE-BASED GENERATIVE MODELS

We use the score-based models originally introduced in Song & Ermon (2019). The score of a probability density function $p_X\left(x\right)$ is defined as $\nabla \log p_X\left(x\right)$. The work in Song & Ermon (2019) proposes to train a noise-conditional score model $s_\theta\left(x;\sigma\right)$ that learns the score of the perturbed distribution $p_{X_i'}$, where $X_i' = X + Z_i$, and $Z_i \sim \mathcal{N}\left(0, \sigma_i^2 \mathbf{I}\right)$, conditioned on $\sigma_i$. In practice, a single deep neural network $s_\theta$ is trained with a weighted combination of denoising score matching losses

at multiple levels (Vincent, 2011; Song & Ermon, 2019):

$$L_{DSM} = \frac{1}{2B} \sum_{i=1}^{B} \sigma_i^2 \left\| s_\theta\left(x_i'; \sigma_i\right) - \nabla \log p_{X'|X}\left(x_i'|x_i\right) \right\|_2^2, \tag{6}$$

where $\sigma_i$ is a noise level chosen at random from a discrete set of exponentially noise levels $(\sigma_i)_{i=1\ldots L}$, where $L$ and $\sigma_1$ are hyper-parameters. We have chosen to use this formulation of score-based models in favour of more recent ones (Song et al., 2021) due to the improvements in Song & Ermon (2020), which we have followed and successfully used as-is, leading to almost zero tuning of the architecture or learning objective required. When using the implementation $s_\theta(x; \sigma) = s_\theta(x)/\sigma$ and the closed-form expression of the gradient in equation 6, we obtain the loss function in equation 2.

If a training set of clean channel matrices $H_i$ is available, the loss in equation 2 is used to train the score-based model; otherwise, when a corrupted database of channels is available, we use the loss in equation 4 to train a denoiser $g_\theta$, which is then converted to a score model as described in equation 2.3. Both of these objectives lead to a learned score model for wireless channels, and this is a separate optimization problem from channel estimation itself, which is done through the iterative process in equation 3.

We use the NCSNv2 architecture from Song & Ermon (2020), which is a RefineNet (Lin et al., 2017) with a depth of four hidden layers and variable number of channels, that doubles in the encoder after the first hidden layer, and is mirrored in the encoder. For Section 2.3, we implement $g_\theta(\boldsymbol{H}; \sigma_i)$ as the same RefineNet and a forward implementation of $\sqrt{1 + \sigma_n^2} g_\theta(\boldsymbol{H}/\sqrt{1 + \sigma_n^2})$ to work with normalized inputs.

We train our models using a training set of 10000 channels from the CDL-C model, as described above, and use the Adam optimizer with a learning rate of $10^{-4}$, batch size of 32 and exponential moving averaging of the weights with a factor of 0.99. Figure 4 shows ablation results on the downstream channel estimation performance when the number of channels in the first layer $N_c$ is varied. In the case of the *Mixed* score-based model, we use 10000 training channels from each of the CDL-B, CDL-C, and CDL-D environment models.

Overall, the estimation performance is very robust to the network size and using a network with $N_c = 6$ only loses up to 1 dB in estimation performance in the high SNR regime. Similar trends are valid across all SNR points. The same conclusion also holds for the generalization to the CDL-D environment (dashed lines in Figure 4).

To determine the best inference hyper-parameters using the in-distribution validation data, we let $\gamma_t = \gamma_0 \eta_t$, and grid search for $\gamma_0$ in the set $\{1, 2, 3, 4\} \times 10^{-10}$, while using an exponentially decaying $\eta_t$ as in Song & Ermon (2020). We additionally search for $\beta$ in the set $\{10^{-3}, 10^{-2}, 10^{-1}, 1\}$. Since we assume the SNR is known for the estimation task, we find a set of hyper-parameters for each SNR value. We generally find that $\gamma_0 = 10^{-3}$ is the best value, regardless of SNR, while $\beta$ decreases as the SNR increases.

## C   DETAILS ABOUT BASELINES

The Lasso baseline solves channel estimation by solving the optimization problem $\arg\min_{\boldsymbol{A}} \frac{1}{2} \|\boldsymbol{Y} - \boldsymbol{F}_r \boldsymbol{A} \boldsymbol{F}_t \boldsymbol{P}\|_F^2 + \lambda \|\text{vec}(\boldsymbol{A})\|_1$ and then using $\boldsymbol{H}^\star = \boldsymbol{F}_r \boldsymbol{A}^\star \boldsymbol{F}_t$ as the output solution. In the previous $\boldsymbol{F}_r$ and $\boldsymbol{F}_t$ are left and right-sided 2D-DFT matrices, respectively of the appropriate sizes. That is, the approach assumes that the channel is sparse in the 2D-DFT representation. We choose the optimal $\lambda$ for each SNR value and sparsity level $\alpha$ by searching in the set $\{0.001, 0.003, 0.01, 0.03, 0.1, 0.3, 1, 3\}$.

The fsAD baseline is similar to Lasso, except that now sparsity is imposed in an over-sampled angular domain. In the case of uniform linear antenna arrays at both the transmitter and receiver, this amount to using the first $N_r$ and $N_t$ rows and columns, respectively, from the DFT matrices of larger size $\boldsymbol{D}_r$ and $\boldsymbol{D}_t$. The solved optimization problem is thus $\arg\min_{\boldsymbol{A}} \frac{1}{2} \|\boldsymbol{Y} - \boldsymbol{D}_r \boldsymbol{A} \boldsymbol{D}_t \boldsymbol{P}\|_F^2 + \lambda \|\text{vec}(\boldsymbol{A})\|_1$, where the optimization variable now has a dimension of $LN_r \times LN_t$, and $L$ is a lifting (oversampling) factor. In all of our experiments, we use $L = 4$ (note that $L = 1$ recovers Lasso) and we search for $\lambda$ in the same way as for Lasso.

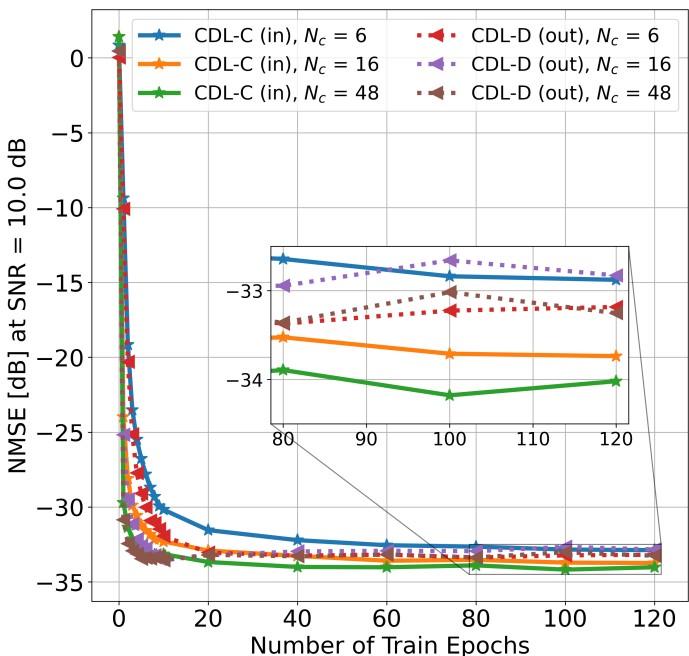

Figure 4: Ablation results for downstream estimation performance as a function of model size and training epochs. Models are trained on CDL-C channels and evaluated at $\alpha = 0.6$, SNR $= 10$ dB.

For the WGAN and GLO baselines, we train a deep generative model paired with a discriminator or a learned latent codebook, respectively. In both cases we use the same basic DC-GAN architecture with three hidden layers, with details available in the source code repository. To solve channel estimation, we use the CSGM framework Bora et al. (2017) for both methods and solve for $\arg\min_{\boldsymbol{z}} \frac{1}{2} \|\boldsymbol{Y} - G(\boldsymbol{z})\boldsymbol{P}\|_F^2 + \frac{\lambda}{2} \|\boldsymbol{z}\|_2^2$, where the final channel estimate is output as $\boldsymbol{H}^\star = G(\boldsymbol{z}^\star)$. We use an Adam optimizer to solve this optimization problem using a pretrained generative model. For both WGAN and GLO, we grid search for the best learning rate among the set $\{0.001, 0.003, 0.01, 0.03, 0.1, 0.3, 1\}$ for each SNR point, as well for the best $\lambda$ in the set $\{0.001, 0.01, 0.1\}$. We generally find that optimization quickly saturates in the high SNR regime after the first 200 steps, regardless of the learning rate and $\lambda$ chosen.

We train the L-DAMP using a U-Net (Ronneberger et al., 2015) backbone with a depth of four and 12 channels after the first hidden layer, that is unrolled for a number of ten steps. To optimize performance, we train a separate L-DAMP model for each SNR value.

