# OpenReview forum: "Score-Based Generative Models for Wireless Channel Modeling and Estimation"
_ICLR.cc/2022/Workshop/DGM4HSD — ICLR 2022 DGM4HSD workshop Poster_

### Official Review · Reviewer_iQvA · 2022-03-19
**Good work fitting the workshops's objectives but lacking some practical details**

**Rating:** 7
**Confidence:** 2

**Review:**

The paper uses score-based models and GANs in multiple-input multiple-output (MIMO) channel estimation. Despite the work seems to be well formulated and evaluated, I have a limited knowledge on the specific application field of wireless channel modelling to be entirely sure about my review. Indeed, most of my comments centre around the fact that I think the paper misses important details to be understood by a wider audience, as I expect this workshop to have.

My specific comments are:
- A main limitation of this paper is that I don't think it fully explains what a "score-based" model is. As this is such a key component of the paper, I believe a few sentences on this are necessary. For example, towards the end of page 2 we see the loss function used but we only know that the score-based model is "s_\theta". What exactly is this, then? Even in Appendix B, this explanation doesn't seem to exist, as the reader is only pointed to some previous model, still leaving my question unanswered, eg. are there other "score-based" models? Where does the name "score" come from in this context? Why is this method specifically chosen by the authors?
- As a suggestion, I think it would be useful to the general audience to have a better explanation of what is a wireless channel in this case, and why it is modelled as a complex-valued matrix. Maybe this could fit in the appendix.
- Before equation 1 on page 2, the authors mention a Gaussian noise "n_i". How realistic is this gaussian assumption for the real-world dataset? Is there any knowledge on whether the noise follows more specific distributions or not really?
- I think the related work section is missing a proper explanation on how this topic was previously tackled in the applied field, and why the focus of the paper is on score-based functions. Indeed, I think the 2nd paragraph of section 1.1 seems to be more of a list of the baselines used in the paper rather than a critical evaluation of the previous literature in which this paper is.

---

### Official Review · Reviewer_Ejo4 · 2022-03-24
**Paper demonstrates potential for score-based generation of wireless channel models well**

**Rating:** 7
**Confidence:** 2

**Review:**

The authors apply a score-based model to the generation of (linear) wireless channel models. In addition to this generative model, they propose a new training objective for the case when only corrupted data with known noise variance is available. In experiments, they demonstrate the quality of generative samples and show that these models are suitable for channel estimation.

This is an interesting topic relevant for the workshop.  While the generative model is mostly an off-the-shelf application of an existing approach to a new data modality, there are novel details and interesting results in this paper.

The method seems solid. The experiments are quite varied and demonstrate clearly that the approach works well (at the cost of computationally expensive sampling, which the authors acknowledge).

The paper is also well-written. Of course space is at a premium in this format, but as someone not very familiar with the field, I would have appreciated more elaborate introductions of the fundamentals. This holds both for channel modelling and channel estimation as well as for score-based / diffusion models. Speaking of score-based generative models: several such models have recently been proposed, it would be interesting to hear more about why the authors chose this particular approach. More broadly, I wish the paper would comment more on the structure of this particular data modality and why the proposed generative model matches that structure well. The conclusions already mention some particularities of the data modality, it would be great if these can be taken into account in future work.

Overall, this is a useful proof-of-concept paper that (with some more space) could benefit from more in-depth discussion. I think it would make a fine fit for this workshop.

---

### Official Review · Reviewer_qG3Q · 2022-03-28
**An interesting application of score-based generative models, but with limited performance.**

**Rating:** 6
**Confidence:** 2

**Review:**

In this work, the authors applied the score-based generative model to wireless channel modeling in order to use the representational power of generative models. However, the advantages and improvements of using the score-based generative model over conventional generative models is not very clear to me.

The explanations of the problem setting and the score-based generative models are not very clear. Although the score-based generative model is developed in another paper, it will be very helpful to briefly explain it, otherwise the $s_\theta$ function is not explained and confusing to readers.

I am a bit confused by the training process. The wireless channel modeling itself is an optimization problem, while the training of the generative models is another optimization problem. I am not clear why optimizing the loss function in Eq 2 solves the score-based generative models for wireless channels.

---

### Decision · Program_Chairs · 2022-03-26

Accept (Poster)